# Role of White Matter Hyperintensities and Related Risk Factors in Vascular Cognitive Impairment: A Review

**DOI:** 10.3390/biom11081102

**Published:** 2021-07-27

**Authors:** Yiyi Chen, Xing Wang, Ling Guan, Yilong Wang

**Affiliations:** 1Department of Neurology, Beijing Tiantan Hospital, Capital Medical University, Beijing 100070, China; cyeeyeee@163.com (Y.C.); wxfx.2005@163.com (X.W.); 2China National Clinical Research Center for Neurological Diseases, Beijing 100070, China; 3Advanced Innovation Center for Human Brain Protection, Capital Medical University, Beijing 100070, China; 4Beijing Key Laboratory of Translational Medicine for Cerebrovascular Disease, Beijing 100070, China; 5Department of Neurology, Chongqing University Central Hospital, Chongqing Emergency Medical Center, Chongqing 400000, China

**Keywords:** cerebral small vessel disease, white matter hyperintensities, cognitive impairment

## Abstract

White matter hyperintensities (WMHs) of presumed vascular origin are one of the imaging markers of cerebral small-vessel disease, which is prevalent in older individuals and closely associated with the occurrence and development of cognitive impairment. The heterogeneous nature of the imaging manifestations of WMHs creates difficulties for early detection and diagnosis of vascular cognitive impairment (VCI) associated with WMHs. Because the underlying pathological processes and biomarkers of WMHs and their development in cognitive impairment remain uncertain, progress in prevention and treatment is lagging. For this reason, this paper reviews the status of research on the features of WMHs related to VCI, as well as mediators associated with both WMHs and VCI, and summarizes potential treatment strategies for the prevention and intervention in WMHs associated with VCI.

## 1. Introduction

Cerebral small vessel disease (CSVD) is a very common type of age-related cerebrovascular disease seen in clinical practice [1]. Characteristic imaging markers of CSVD—white matter hyperintensities (WMHs) [2]—are prevalent in older adults [3]. The prevalence of WMHs has been observed at 25.9% in young clinical outpatients [1] aged below 45, increasing to 50% in community-dwelling adults aged 44–48 [2] and to 60–100% in individuals over 65 [4,5]. The Rotterdam scan study found a 0.2% increase per year in the prevalence of subcortical WMHs, and a 0.4% increase per year in periventricular WMHs. In participants aged 60–70 years, 87% had subcortical WMHs and 68% had periventricular WMHs; among those aged 80–90 years, 100% had subcortical and 95% had periventricular WMHs [5,6]. In the WMH population, the incidence rates of stroke, dementia, and mortality are 2.6–4.4×, 1.3–2.8×, and 1.6–2.7× those of the normal population, respectively [7,8,9,10].

In 2011, the American Heart Association/American Stroke Association defined the concept of vascular cognitive impairment (VCI) as a series of syndromes caused by cerebrovascular disease and its risk factors. The concept VCI embodies all types of cognitive impairment, vascular dementia being the most severe VCI and mild VCI being the initial stage of VCI [11]. In a meta-analysis of 22 longitudinal studies, WMHs were found to be significantly associated with occurrence of cognitive impairment, with a 2× increased risk of dementia in WMH patients, and a higher risk of cognitive impairment was associated with a higher WMH load [7]. The LADIS study also found that WMHs were related to the yearly rate of decline in clinical memory, attention, executive function, and global cognition [12]. In the early stages of cognitive impairment, changes are observed in the intensity of white matter and in the cerebrospinal fluid levels of amyloid beta (Aβ_40/42_) [13,14]. Because the baseline severity of WMHs is considered an independent predictor of clinical VCI [15], some scholars support the use of WMHs in research as an intermediate marker of cognitive impairment, while clinically they support the detailed screening of those with WMHs for the risk factors for VCI. As outlined above, WMHs are closely associated with the occurrence and development of VCI, and are regarded as a risk factor and independent predictor of VCI. Moreover, they may appear prior to clinical symptoms of cognitive impairment.

In view of this, our review herein aims to summarize: (1) the features of WMHs associated with VCI; (2) several mediators of the association between WMHs and VCI; and (3) plausible therapeutic strategies for VCI with WMHs.

## 2. Relationship between WMHs and Cognitive Impairment 

VCI-related tissue damage is mostly observed in white matter, less so in gray matter. The histopathology of white matter injury encompasses focal and diffuse lesions, most commonly presenting in combination in various proportions [16]. A focal lesion is regarded as the result of an acute reaction to regional ischemia, whereas a diffuse white matter lesion is considered an adaptation to abnormal perfusion and physiological conditions within the tissue. Moreover, focal changes may progressively impair the surrounding normal tissues, which would ultimately result in a diffuse abnormality more than 200 times the volume of the original injury, known as a lacunar infarct [16]. Thus, white matter changes may in part reflect a progressive pathology of normal white matter responding to hemodynamic alterations. Importantly, WMHs may be a late-stage outcome of such white matter changes [17]. 

Fluid attenuated inversion recovery sequences may insufficiently unravel the heterogeneity of WMHs in terms of histopathology. The histological basis of WMHs has been suggested to include diffuse myelin sparse, retaining the subcortical U-shaped fibers, astrocyte hyperplasia, cavernous cell hyperplasia, axonal loss, and enlarged perivascular spaces [18,19]. Nevertheless, autopsy studies have shown that white matter injuries may present variously, from mildly untangling of matrix to different degrees of myelin and axonal loss [20]. Although the insensitivity of imaging methods may attenuate the relation between magnetic resonance imaging (MRI) abnormalities and the severity of clinical symptoms, imaging studies have nevertheless found a relation between cognitive impairment and WMHs, mainly involving the severity of the WMHs and their location.

### 2.1. White-Matter Lesions: Severity

Measurements of the severity of WMHs is mostly based on WMH volume or a semiquantitative visual grade (the most common being the Fazekas visual grade and the Schelten’s grade). Multiple population-based studies have determined the correlation between WMH volume and whole-brain cognitive decline. Cross-sectional studies suggest that a larger volume of WMHs, accompanied by higher Fazekas visual grades, associates with lower functioning of the entire brain or with specific regional cognitive performance deficits, but these studies show a weak evidential effect [7,21,22]. However, current research reveals that WMHs are particularly relevant to decreased information processing speed and executive function [23].

Nevertheless, the visual rating scale has a ceiling effect and cannot accurately reflect WMH changes. The progression of WMHs, rather than their baseline level, is significantly correlated with cognitive decline, and this provides strong evidence for a causal relationship [24,25]. Therefore, automatic or semi-automatic analyses of WMH variation might better predict cognitive impairment. These augmentation methods obviously reduce intra- and inter-rater variation and work-time efficiency in quantifying WMH lesion burden and progression through the use of image algorithms [20]. Interestingly, clinical observations show that cognitive function in patients with the same WMH volume or visual rating score could be either impaired or preserved. Furthermore, all the abovementioned clinical imaging studies were insensitive to the pathophysiological process and did not elucidate the connection between WMHs or their development and VCI. Consequently, functional magnetic resonance was expected to better characterize the anatomy and functional networks of WMHs and to better evaluate VCI risk.

### 2.2. White-Matter Lesions: Location

VCI is believed to result from interruption of cortico–subcortical and cortico–cortical connections secondarily to diffuse white matter injury, multiple lacunae, and microbleeds [15]. It is believed that WMHs affect brain tissue beyond the recognized focal lesions, inducing a series of reactions that propagate from local lesions to remote areas of normal brain. This effect probably accounts for the inconsistent clinical presentations following WMH onset. On one hand, WMHs can cause atrophy in distal, corresponding cerebral cortex via damage to long-range axons that traverse the brain in white matter tracts [26] (Figure 1). On the other, WMHs and hippocampal atrophy observed on MRI have shown an additive effect, and periventricular WMHs are correlated with hippocampal atrophy [27,28]. Taken together, these results might explain the effect of lesion location on the deterioration of cognition.

Moreover, WMHs at specific locations have been found to be better related to cognitive impairment than WMH volume, particularly impairment of executive function and episodic memory. The anatomic locations closely associated with the potential of WMHs to induce cognitive decline include the thalamus, striatum, and connections between the frontal or parietal lobes and the occipital lobe [30]. One study of stroke patients showed that frontal WMHs are associated with executive dysfunction, while temporal WMHs are associated with memory disorder. WMHs in the left dorsolateral prefrontal cortex are related to decreased performance in a working memory task. Lesions along the cholinergic pathway have also been shown to be related to memory impairment. WMHs indicate damage to the connecting nodes within the brain’s functional network, which leads to disorders of the sub-frontal cortical pathway, including conduction by the cholinergic white matter bundle, as well as reduction of the long-range connections among three major attention networks: the default mode network (DMN), the frontoparietal control network (FPCN), and the dorsal attention network (DAN), along with their hubs (Figure 1) [29,31].

With the deepening of understanding, a theory of damage to white-matter integrity emerges based on WMH results, which reflect damage to both anatomical structures and functional networks. Loss of white-matter integrity often appears in normal-appearing white matter (NAWM) near WMHs prior to becoming evident on MRI, and related to VCI [32,33]. Recently, nuclear resonance diffusion tensor imaging (DTI) has been used to evaluate white matter integrity and its parameters, which are associated with the severity of CSVD in a graded manner. Brain fractional anisotropy and mean diffusibility have proved to be sensitive indicators of Mini-Mental State Examination (MMSE) and executive function outcomes [34]. Moreover, DTI can map neuronal circuits for cognition. For example, the Papez circuit connects the anterior thalamic nucleus, mammillary body, and hippocampus, and is composed of the superior longitudinal fascicle, fornix, and mammillothalamic tract; all of these are in the periventricular white matter and are vulnerable to periventricular white-matter lesions [34].

Most longitudinal studies suggest that the relationship between WMHs and cognitive impairment is more pronounced in periventricular regions than in non-periventricular, which might be due to periventricular fibers implementing long-range association transmission, inter-hemispheric transmission, and long projections generally, while deep white matter transmission involves relatively short association fibers in specific encephalic regions. In terms of the anterior-posterior axis, anterior white matter lesions may be associated with decreased processing speeds, while in posterior areas they may be associated with impaired visual-constructional function [35,36]. In addition, the spatial distribution of other imaging biomarkers of CSVD, including lacunar infarcts and cerebral microbleeds, could also result in various neuropsychological symptoms and should be noted. For example, those located at the basal ganglia or thalamus possibly disrupted the connectivity to the prefrontal cortex and resulted in executive dysfunction or impaired processing speed [37,38]. A longitudinal study with an 18-month follow-up period compared first-ever lacunar stroke patients with and without mild VCI and showed that significant WMH changes during intervals were observed in both groups; however, at baseline, patients with mild VCI experienced marked progressive gray matter atrophy in the frontal and temporal cortices and subcortical regions, such as pons, caudate, and cerebellum, compared with those without VCI [39]. The signs of cognitive deficit are proposed to be only significant in patients with severe disruption of white matter integrity; these patients are thought to exhibit a “threshold effect,” and such a non-linear relationship between the integrity of white matter and WMHs can at least partly explain the presence of extensive WMHs in patients without symptoms. In view of this, when considering the influence of topographical distributions of lesions on increased risk of VCI, the extent of neuroimaging markers of CSVD (i.e., the existing WMHs, lacunar infarcts, cerebral microbleeds, or atrophy) should be considered all together in evaluation.

## 3. Risk Factors Linking WMHs to VCI

### 3.1. Blood-Brain-Barrier (BBB) Function

Pathological studies suggest that BBB dysfunction and interstitial fluid accumulation around cerebral blood vessels, especially small arteries, might represent the early pathological changes that lead to WMHs. Dynamic contrast-enhanced magnetic resonance imaging (DCE-MRI) is therefore widely used as an indicator of WMHs, as well as for assessing the functional integrity of the BBB [40]. Additionally, many WMH studies have confirmed that BBB damage is related to the development of VCI [41]. Previous studies have found that substances in the blood of patients with WMHs, such as fibrinogen and immunoglobulin, can enter the brain parenchyma via capillaries, which led to the insight that patients with WMHs have BBB dysfunction or damage. Further, plasma components leak into intervascular and perivascular tissues through a dysfunctional BBB or through damaged endothelial cells in perforating arteries to cause neuronal damage. This is regarded as one of the pathogenic mechanisms of WMHs. Other studies suggest that BBB permeability increases with normal aging, which itself may be an important mechanism in the initiation or worsening of CSVD, an age-related disease. A cohort study concluded that BBB dysfunction is the core mechanism behind CSVD and dementia, given that leakage of the BBB in the regions of WMHs was increased and predicted future cognitive deterioration [40,41,42,43].

Although we believe that the BBB exerts a crucial role in the development of WMHs, the molecular mechanism thereof remains to be unraveled. Mounting evidence suggests that BBB disruption leads to circulating neurotoxic molecules entering the brain, compromised clearance of neurotoxic molecules (e.g., increased amyloid precursor protein expression and Aβ clearance disorder), abnormal energy metabolites and nutrient delivery, and aberrant expression of growth factors, matrix molecules, pro-inflammatory mediators, and vascular receptors, all of which may result in synaptic and nerve-ending dysfunction, finally influencing cognitive functions [44,45]. Multiple studies on rats fed with high energy diets, especially diets rich in saturated fats and cholesterol, have found increased BBB permeability along with cognitive dysfunction, indicating a link between up-regulated lipid metabolism, the BBB, and cognitive function [46,47].

To date, advanced imaging technologies that detect neural metabolites, metabolic states, or molecular profiles in the brain include fluoro-deoxyglucose positron emission tomography (FDG-PET), P-glycoprotein (P-gp)-PET, amyloid β-protein (Aβ)-PET, microtubule-associated protein tau-PET, and proton MR spectroscopy (^1^H-MRS). These are powerful imaging tools, and the combination of imaging techniques and biological processes is expected to help us better understand the pathophysiology of WMH and its involvement with VCI [44]. For example, ^1^H-MRS, which measures the neuronal metabolites creatine and N-acetylaspartyl compounds in white matter, is a more reliable determinant of the cognitive consequences of VCI pathology than is white-matter lesion load [48]. Additionally, radioactive iodized tracers have been used to show fluid, soluble metabolites (including soluble Aβ), and tracers flowing along the cerebral artery wall, which, although currently used mainly in animal experiments, may be applied to clarify the pathophysiological process of cognitive impairment associated with WMHs [49].

### 3.2. Inflammatory Biomarkers

Neuroinflammation contributes to white-matter damage, which further promotes cognitive dysfunction. Mounting evidence indicates that neuroinflammation plays a key role in the process of cognitive decline. Systemic inflammatory markers such as C-reactive protein, tumor necrosis factor-alpha (TNF-a), and interleukin-6 (IL-6) are all implicated in cognitive decline [50]. Hypothesized mechanisms have advanced to explain neuroinflammation-induced cognitive decline, including signal transduction by the Receptor for Advanced Glycation End Products (RAGE). Activation of RAGE, which is expressed on microglia and neurons in the central nervous system, including the hippocampus, entorhinal cortex, and superior frontal gyrus, enhances nuclear factor-kappa B (NF-κB)-dependent transcription, which is a key regulator of proinflammatory cytokine release, and might ultimately cause brain damage and affect cognitive ability [51]. In addition, Aβ peptide activates cyclooxygenase-2 (COX-2) dose-dependently, and treatment with COX-2 inhibitors or ibuprofen have showed preventive effects that indicate non-steroidal anti-inflammatory drugs may be effective for lessening the risk of dementia [52]. Previous studies have shown that high expression of Tumor Necrosis Factor-α (TNF-α) and interferon-γ (IFN-γ) activates inducible nitric oxide synthase, which mediates inflammation and endothelial damage in cerebral microvascular structures, which would then appear as WMH on MRI and eventually lead to the progression of VCI [53].

### 3.3. Cerebral Aβ and Tau Burden

Unlike the previously mentioned small-vessel-disease-related WMHs, which are associated with spared subcortical U-shaped fibers, WMHs have extended to include the U-fibers in a PSEN1 mutation carrier, a genetic form of Alzheimer’s disease (AD) [54]. Aβ dysmetabolism has been associated with reduced microstructural integrity and WMHs, effects limited to WMH areas and not observed in NAWM regions [55]. Specifically, the association has been found between cerebral amyloid burden and WMHs located in the posterior periventricular regions and the splenium of the corpus callosum [56]. Also, in those with cognitive impairment but not dementia, a possible synergistic effect has been observed between cerebral amyloid burden and WMH in cases of worsening cognition [57]. However, whether amyloid pathology shows a causal link with WMH progression and is associated with eventual VCI remains unclear due to lack of data, although tau protein has been implicated in the association of WMHs with impairment of cognition. Increased tau, measured by AV1451 uptake, has been preferentially elevated in bilateral inferior temporal brain regions and independently associated with CSVD burden (including WMHs, lacunas, and microbleeds) regardless of Aβ levels. Meanwhile, tau is presently considered the final common pathway by which either Aβ or WMHs leads to VCI. It should be noted that in patients with VCI, impaired cognitive functions associated with tau have been demonstrated in language and general cognition rather than in memory, which has been most affected in AD [58]. These results raise the question of whether tau accumulation may be differentiated from its role in AD spectrum disorders in relation to WMHs and VCI. It is possible that accumulated tau contributes to VCI either by triggering continuous neurotoxic processes and vascular damage including WMHs, or by its local cumulative effects on the interruption of the cholinergic pathway [59] in combination with the microstructural alterations of WMHs [60]. Recent findings have shown that cells in the CNS can release extracellular vesicles (EVs), which are a series of tiny double-membrane structures (20–2000 nm) with membrane-related interaction regions [61], filled with different molecular materials (including proteins, lipids, DNA, and various RNA species). EVs target nerve cells and blood vessel cells through a very dynamic and adaptable cell-cell communication mechanism, and change the function of cells through the delivery of substances, thereby participating in the process of mediating neurovascular regeneration and remodeling, anti-apoptosis, and anti-inflammation and other processes [62]. Increased evidence has shown that specific proteome changes in EVs, and these changes in the biogenic origin of EVs are closely related to the occurrence and development of cognitive impairment, blood perfusion, and cell apoptosis [63,64]. However, their roles in the occurrence and development of WMHs, as well as their biomarker potential in predicting VCI risks or as a therapeutic approach for VCI in WMH patients, are promising and should be further examined.

### 3.4. Metabolic Abnormalities

Insulin receptors are highly expressed in cognition-related regions of the brain, as well as within the BBB. In mouse models of diabetes, decreased brain insulin signaling raised the levels of tau phosphorylation and of Aβ peptide, both of which are biomarkers of cognitive impairment. Of note, metabolic changes in diabetes affect white matter tissue, in line with the observation that persons with diabetes have a higher prevalence of WMHs and cognitive decline than do those without [65]. Hyperglycemia increases the production of reactive oxygen species [66] and inflammatory cytokines [67], which induces endothelial impairment and is associated with higher WMH volumes [68]. In rats with impaired fasting glycemia, white-matter lesions developed prior to type-2 diabetes mellitus (DM) onset [69]. In a type-2 DM animal model, white matter integrity was disrupted in a time-dependent manner, possibly due to injured oligodendrocytes and the resulting demyelination [69]. DM and prediabetes both associated with the white-matter connection network [70,71,72]. Therefore, altered insulin signaling or abnormal glucose metabolism may be a factor in cognitive impairment and involve the development and progression of WMHs, but more research is needed to decode how white matter injury and cognitive decline are directly linked to insulin resistance and hyperglycemia.

Obesity is considered an independent risk factor for worsening cognition [73]. After bariatric surgery, a study found significant improvements in memory function 24 months postoperatively compared with obese controls [74]. Compared to non-obese people, obesity is associated with a reduction in total brain volume and white matter integrity [75]. Mouse model studies have shown that diet-induced obesity raises the level of reactive oxygen species in the brain, and cognitive dysfunction following the development of obesity may be caused by increased oxidative stress [76]. Moreover, patients with metabolic syndrome have a higher brain fatty acid intake and more fatty acid accumulation in white matter than in other brain regions, with white matter showing the highest average percent increase [75]. Lipid oxide variation consistent with higher soluble epoxide hydrolase activity is considered a marker of VCI, a correlation that can partly be explained by the observed periventricular subcortical white matter damage [77]. The precise mechanism of deep or periventricular WMH formation exhibits some heterogeneity [78]. A population-based cohort study showed that a greater waist-to-hip ratio (WHR, an indicator of visceral fat) was independently related predominantly to deep white matter burden, and mediation analyses suggested that this association was via a raised level of proinflammatory cytokines [79]. Using data from UK Biobank, a functional MRI study revealed interaction effects between the burden of periventricular WMHs in obesity (grouped by WHR) and functional connectivity in the orbitofrontal cortex, which controls inhibitory behavior [80]. Others than that, a mediation effect of WHR was found between processing speed and total WMH burden [81]. The total WMH burden and the amount of WMHs in deep or periventricular brain regions are not independent. These studies collectively suggest a detrimental effect of both WMHs and obesity on cognition. These inconsistent results may provide a rationale for exploring the link between the spatial distribution of WMHs and cognition in persons with obesity.

### 3.5. Other Vascular Risk Factors for WMHs-Related Cognitive Deficit

The spontaneously hypertensive, stroke-prone rat (SHRSP) is an animal model that exhibits essential hypertension, stroke and features of the histological changes of CSVD, including endothelial damage and BBB leakage [82]. The learning/memory dysfunction observed in SHRSP can be partly attributed to an impaired cerebral neurotransmitter system resulting in reduced acetylcholine and choline levels in the cortex, as well as to thrombosis resulting from structural changes (blood components and blood velocity in vessels) resulting from prolonged hypertension [83]. Although with complex confounding variables, especially aging, hypertension and increased arterial stiffness and impaired WMH microstructure interact in a vicious cycle [84,85]. In hypertensive elderly persons, WMH volumes increased in both normotensive controls and hypertensive groups at almost two years follow-up, and a significant linear trend in WMHs volume progression was found, being highest in the hypertensive group, followed by the treated hypertensive group, and lowest in the normotensive group [86]. In hypertension, hypoperfusion in either whole brain or discrete brain regions was found, adding to the complexity of hemodynamic variation over a lifetime [87]. Hypoperfusion is one of the etiologies of VCI [88]. At midlife (40–65 years), hypertension is significantly associated with an increased risk of both late-life cognitive impairment and dementia [89,90], WMH progression [91], and disrupted white-matter integrity [92]. It is therefore plausible that hypoperfusion is a potential culprit in later phases. The association during aging between blood pressure profiles and WMHs, and their involvement in VCI, remains in need of more insight and better definition.

Additionally, several lines of evidence suggest that plasma homocysteine (Hcy) concentrations strongly associate with WMHs in a dose-dependent manner. Hcy-lowering therapy has significantly slowed the progression of WMHs, and Hcy can lead to alterations within the blood vessel wall. The main mechanisms responsible for these modifications include direct endothelial damage attributed to increased oxidative stress and proinflammatory effects [93,94,95]. Hyperhomocysteinemia is correlated with cognitive impairment, especially in frontal-executive function [96]. In elderly persons with WMHs, positive correlations were found between elevated plasma Hcy levels and the severity of WMHs and cognitive impairment [97].

Finally, local factors within the vasculature such as shear stress or platelet function are vital to vascular homeostasis. Hemodynamic changes such as low carotid wall shear stress (specifically, diastolic wall shear stress) are associated with WMHs and cognitive impairment [98,99]. White-matter lesions accompanied by cognitive decline are also considered to be linked to activation of platelet function [100]. In contrast to these factors, diagonal ear lobe crease (DELC) has been proposed as a marker for atherosclerotic disease [101,102]. Interestingly, DELC was more frequently found in persons with VCI than in cognition-normal persons. In those with cognition impairment, the presence of DELC may predict a greater WMHs burden and Aβ positivity [103].

## 4. Treatment of Cognitive Impairment Associated with WMHs

This can be divided into primary prevention and secondary prevention. The primary prevention strategy aims to maintain factors of health and eliminate WMH-associated risk factors before cognitive symptoms occur. The secondary prevention strategy begins after cognitive or emotional symptoms become apparent, with the goal of preventing further deterioration to dementia [20]. 

Because the severity of WMHs correlates with the development of VCI, treatment that aims to attenuate WMH progression to prevent the occurrence or development of VCI seems reasonable. Instead of non-modifiable risk factors such as age, sex and genetic variations, the manageable risk factors for WMHs including blood pressure variability, DM, dyslipidemia, smoking, and obesity should also be valued as controllable components for VCI. 

Although a strict target (systolic pressure < 120 mm Hg) provides benefits in slowing WMH progression, studies with such a focus have shown inconsistent effects on cognition [104,105,106]. Therefore, expert consensus advocates that blood pressure control should be initiated at >140/90 mm Hg and that a tight treatment goal could be considered together with careful side-effect evaluation and monitoring for cerebral hypoperfusion in middle-aged and older adults with vascular risk factors. Importantly, in patients with concomitant cerebral vascular disease, the intensity of therapy should be further reevaluated [107,108]. Youth onset [109] and midlife onset [110] of type-2 DM have both been associated with higher risk of cognitive impairment, especially in executive function, working memory, and global cognitive score, possibly due to the loss of hippocampal and whole-brain volume [111]. However, convincing data that glycemic control reduces the risk of dementia or cognitive decline is lacking [112]. Impaired glucose metabolism has accelerated brain aging compared with normal, from 2.3 to 10.4 years [70]. Meanwhile, considering glycemic control protects multiple target organs (e.g., eyes and kidney), active intervention against dysglycemia should focus on the preservation of brain networks and cognitive function.

Metabolic syndrome is a cluster of abnormalities that includes hypertension, insulin resistance, abdominal obesity, and dyslipidemia [113]. In addition to elevated blood pressure and hyperglycemia, both abdominal obesity and dyslipidemia have shown associations with WMHs and VCI. Randomized, controlled clinical trials of lipid-lowering therapy for the prevention of cognitive decline has so far been unsatisfactory [114,115]. Nevertheless, higher low-density lipoprotein levels have been found to be associated with greater cerebral amyloid and tau deposition [116], which further aggravates WMHs and is associated with VCI. Patients initiated with statins were determined more likely to have other risk factors for dementia, such as APOE-ε4 carrier status or cardiovascular comorbidities. Both the APOE-ε4 allele and hyperlipidemia were significantly associated with a marked WMH progression [117,118]. Although there is no consensus on the effect of hyperlipidemia on WMHs [119,120,121], the results of sustained use of statins as part of a holistic approach instead of statin use alone have suggested a potential decreased 10-year risk of dementia or death [122]. A systematic review identified 13 longitudinal studies and 7 randomized, controlled trials, concluding that weight loss (as diet, physical activity, or bariatric surgery) in overweight individuals is associated with various extents of improvement in attention, memory, or executive function [123]. As for physical exercise, aerobic activity of moderate intensity (50% of VO_2_ max) or longer duration appears to slow WMH progression in persons with VCI [124]. Resistance training was also found to reduce WMH progression in persons with or without cognitive impairment [125,126]. Physical exercise influences cognitive impairment and dementia, and the type of training and its duration and frequency have varied effects on each specific cognitive domain (i.e., executive function [127], selective attention or response inhibition [128], and global cognitive performance [129]). While experimental support remains low for the recommendation of physical exercise as an intervention for reducing the risk of cognitive decline in CSVD patients [130], maintenance of a healthy lifestyle and avoidance of overweight are still reasonable recommendations. Furthermore, isolated executive dysfunction could be observed in the early phases of VCI [38]. Impairment in some neuropsychological tests such as semantic verbal fluency and short delayed verbal memory could possibly exert as initial identifiable abnormalities in CSVD for future development of VCI [131]. Therefore, the development of a comprehensive and robust battery of neuropsychological tests covering cognitive domains including, but not limited to, visuospatial function, executive function, memory, motor skill development, and verbal ability should be attempted in prospective studies to further confirm the neuropsychological tests’ value as a powerful tool for the early detection of future VCI in elderly individuals with a silent cerebral vascular disease, such as CSVD.

In terms of secondary prevention, clinical trials of drugs like cholinesterase inhibitors, central nicotinic receptor modulators, and N-methyl-d-aspartate receptor antagonists have shown some effect on vascular cognitive impairment caused by WMHs. In addition, studies have also shown that remote ischemic conditioning appears to have potential effects on patients with CSVD in terms of improving WMH outcome, cognition, cerebral perfusion, and vascular risk factors. This amelioration may be related to myelin regeneration and reconstruction of the cerebral network [132,133,134]. However, whether attenuating the progression of VCI is an effective treatment strategy for preventing dementia, as well as the best time to start and the most effective secondary interventions for patients with VCI, remain subjects of debate. Considering their heterogeneous nature, the coexistence of WMHs further complicates the selection of appropriate treatments for VCI.

## 5. Conclusions

WMHs are considered an important factor in the occurrence and progression of cognitive impairment. WMHs detected with conventional MRI show heterogeneity in pathology for which we lack a systematic method of evaluation considering both white matter microstructural integrity and functional alterations, and this increases the difficulty of achieving a strong, uniform predictive capability for VCI. Therefore, advanced neuroimaging methods that can determine the possible pathological processes underlying WMHs have been developed, and once analyzed in combination with longitudinal data having cognitive endpoints, will altogether help us identify the mechanisms that underlie WMH formation and VCI. It seems that the mechanism by which WMHs participate in eventual cognitive deficits could be driven by coexisting neurodegenerative and vascular processes, or by either pathway. Since an effective therapeutic approach to WMH reduction for the prevention of VCI remains elusive, its management implies comprehensive vascular risk-factor control and lifestyle modification. Clinical trials should be encouraged in the fields of BBB protection, oxidative stress, inflammation, amyloid dysmetabolism, and others for the identification of novel treatment targets indicated by WMH observations to reduce the risk of VCI.

## Figures and Tables

**Figure 1 biomolecules-11-01102-f001:**
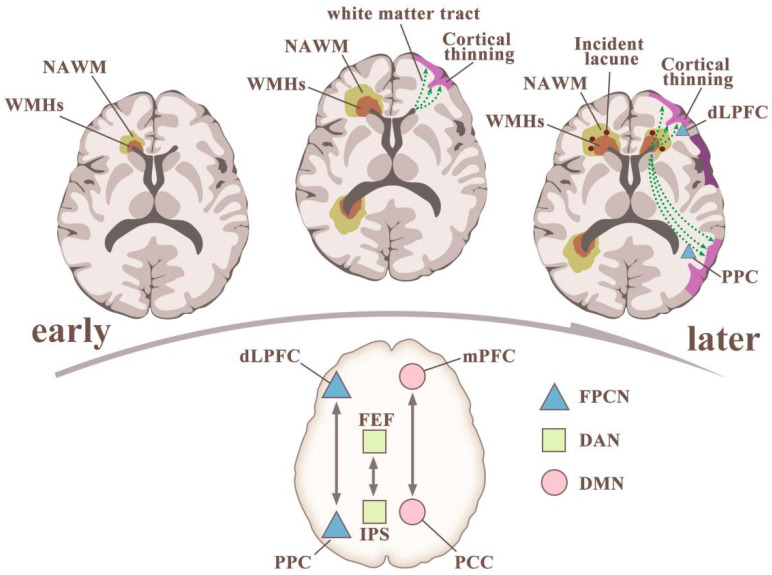
The association between white-matter hyperintensities (WMHs) and cognitive impairment: insights from imaging. Upper left: Interstitial fluid increase leads to WMHs and affects the normal-appearing white matter (NAWM) around it. Patients with WMHs have greater blood-brain-barrier leakage than do controls in the WMHs, NAWM, and cortical gray matter. Upper middle: WMHs progress, leading to secondary cortical thinning and possibly white-matter tract lesions in the NAWM and WMHs. Upper right: WMHs are present, incident lacunae might appear, secondary cortex is thinned, long-tract degeneration worsens, and major attentional networks and their hubs have been injured. (4) The three commonly studied attention networks and their hubs are interrelated. The default-mode network (DMN) includes the medial prefrontal cortex (mPFC) and posterior cingulate cortex (PCC) as major hubs. The frontoparietal control network (FPCN) includes the dorsolateral prefrontal cortex (dLPFC) and the posterior parietal cortex (PPC) as major hubs. The dorsal attentional network (DAN) includes the frontal eye fields (FEF) and the area around the intraparietal sulcus (IPS) as major hubs [29].

## Data Availability

Not applicable.

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
