# Peer review of "Role of White Matter Hyperintensities and Related Risk Factors in Vascular Cognitive Impairment: A Review"

_biomolecules, 2021, doi:10.3390/biom11081102_

Round 1
Reviewer 1 Report
The authors present a comprehensive and updated review of the role of white matter hyperintensities (WMHs) in cognitive decline. The review of the literature is adequate. The following considerations may help to improve the quality of the report:
1. Authors should clearly reflect in the text that mild cognitive impairment of vascular etiology in cerebral small vessel disease is often related to WMHs, but also to silent lacunar infarcts, microbleeds, and cerebral atrophy shown on the neuroimaging. Also, since cognitive impairment is an essential clinical feature of Binswanger's disease, -an historical entity with a neuropsychological characteristics of subcortical vascular dementia-, I would suggest expanding on this aspect in the Introduction (Expert Rev Neurother 2009; 9: 1201-1217). Add and comment on the reference.
2. The authors clearly state the relevance of cerebral atrophy in WMHs. It would be interesting to emphasize in the text and as a new emerging feature of cerebral small vessel disease, the progressive atrophy of grey matter, documented in patients with stroke of the lacunar type (see and comment on the study published in Cerebrovasc Dis 2010; 30: 157-166).
3. In the text, authors should comment that cerebral small vessel disease is the most frequent silent cerebrovascular disease. Likewise, a clinical study reported that half of the patients with a first-ever lacunar infarct present mild cognitive impairment with subcortical vascular features and its presence may be a predictor of subcortical vascular dementia in the medium to long term (see data and comment on a study published in J Neurol Sci 2007; 257: 160-165).
4. It would be interesting for the authors to clearly emphasize the need for future clinical trials in lacunar stroke patients to also include a “neuropsychological study”, because more than half of patients with a first-ever lacunar stroke and without cognitive impairment present minor neuropsychological alterations. These minor disturbances are mainly related to the presence of clinically silent lacunar infarcts, with no relation of cognitive impairment to WMHs at this early stage of cerebral small vessel disease (see and comment on the study published in BMC Neurology 2013; 13: 203). The follow-up of these patients may be interesting to assess whether neuropsychological disturbances are a predictor of subcortical vascular dementia in the medium term.
Author Response
- Authors should clearly reflect in the text that mild cognitive impairment of vascular etiology in cerebral small vessel disease is often related to WMHs, but also to silent lacunar infarcts, microbleeds, and cerebral atrophy shown on the neuroimaging. Also, since cognitive impairment is an essential clinical feature of Binswanger's disease, -an historical entity with a neuropsychological characteristics of subcortical vascular dementia-, I would suggest expanding on this aspect in the Introduction (Expert Rev Neurother 2009; 9: 1201-1217). Add and comment on the reference.
The authors clearly state the relevance of cerebral atrophy in WMHs. It would be interesting to emphasize in the text and as a new emerging feature of cerebral small vessel disease, the progressive atrophy of grey matter, documented in patients with stroke of the lacunar type (see and comment on the study published in Cerebrovasc Dis 2010; 30: 157-166).
Response: We would like to thank the reviewer for evaluating our manuscript and for his/her constructive comments. In 2013, Wardlaw et al. presented the imaging manifestations for the diagnosis of CSVD using standard magnetic resonance imaging (MRI) sequences. They were summarized as the STandards for ReportIng Vascular changes on nEuroimaging (STRIVE) and comprised the following: 1) recent small subcortical infarct, 2) white matter hyperintensity (WMH), 3) lacunae, 4) enlarged perivascular space, 5) cerebral microbleed, and 6) cerebral atrophy. As for Comments 1 and 2, our current review mainly focused on the characteristic imaging markers of CSVD—white matter hyperintensities (WMHs)—which are prevalent in older adults. However, after carefully reviewing the two studies that the reviewer suggested, we considered that besides WMHs, other CSVD imaging markers with specific spatial distribution may also be vital in the contribution of VCI. Thus, we have added the following parts in the “White matter lesions: location” subsection of the revised manuscript:
“In addition, the spatial distribution of other imaging biomarkers of CSVD, including lacunar infarcts and cerebral microbleeds, could also result in various neuropsychological symptoms and should be noted. For example, those located at basal ganglia or thalamus possibly disrupted the connectivity to the prefrontal cortex and resulted in executive dysfunction or impaired processing speed [37,38]. A longitudinal study with an 18-month follow-up period compared first-ever lacunar stroke patients with and without mild VCI and showed that significant WMHs changes during intervals were observed in both groups; however, at baseline, patients with mild VCI experienced marked progressive gray matter atrophy in the frontal and temporal cortices and subcortical regions, such as pons, caudate, and cerebellum, compared with those without VCI [39].” (Lines 176-186)
“In view of this, when considering the influence of topographical distribution of lesions on the increased risks of VCI, the extent of neuroimaging markers of CSVD (i.e., the existing WMHs, lacunar infarcts, cerebral microbleeds, or atrophy) should altogether be considered in evaluation.” (Lines 192-195)
- In the text, authors should comment that cerebral small vessel disease is the most frequent silent cerebrovascular disease. Likewise, a clinical study reported that half of the patients with a first-ever lacunar infarct present mild cognitive impairment with subcortical vascular features and its presence may be a predictor of subcortical vascular dementia in the medium to long term (see data and comment on a study published in J Neurol Sci 2007; 257: 160-165).
4. It would be interesting for the authors to clearly emphasize the need for future clinical trials in lacunar stroke patients to also include a “neuropsychological study”, because more than half of patients with a first-ever lacunar stroke and without cognitive impairment present minor neuropsychological alterations. These minor disturbances are mainly related to the presence of clinically silent lacunar infarcts, with no relation of cognitive impairment to WMHs at this early stage of cerebral small vessel disease (see and comment on the study published in BMC Neurology 2013; 13: 203). The follow-up of these patients may be interesting to assess whether neuropsychological disturbances are a predictor of subcortical vascular dementia in the medium term.
Response: We would like to thank the reviewer for pointing this out. Regarding Comments 3 and 4, we agree that neuropsychological evaluation could possibly be a better prediction tool for the early identification of future VCI in patients with silent cerebral vascular disease, such as CSVD. We have added this information to the “Treatment of cognitive impairment associated with WMHs” subsection of the revised manuscript as follows:
“Furthermore, isolated executive dysfunction could be observed in the early phases of VCI [38]. Impairment in some neuropsychological tests such as semantic verbal fluency and short delayed verbal memory could possibly exert as initial identifiable abnormalities in CSVD for future development of VCI [131]. Therefore, the development of a comprehensive and robust battery of neuropsychological tests covering cognitive domains including, but not limited to, both visuospatial function, executive function, memory, motor skill development, and verbal ability should be attempted in prospective studies to further confirm its value as a powerful tool for the early detection of future VCI in elderly individuals with a silent cerebral vascular disease, such as CSVD.” (Lines 436-444)

Reviewer 2 Report
The manuscript by Chen et al. reviews the state of the art on the investigation of WMHs and their implications in apparition of dementia. As stated by the authors, there is a current lack of available methodologies to identify WMHs in incipient form that may be related to dementia. This reviewer finds this work well suited for the journal and timely regarding its relevant publication. There is a minor issue that should be attended by the authors as follows:
1. Although the main focus of the manuscript is on neuroimaging techniques, the authors include in their work a balanced load of pathophysiology and molecular neuropathology references related to WMHs, lacunar infarctions and dementia. In this regard, this reviewer considers that it would be interesting to add recent knowledge generated on the field of brain extracellular vesicles regarding WMHs and the vascular component of dementia and AD (see the following PUBMED identifiers PMID: 30629763; PMID: 32384937; PMID: 33098484;PMID: 29725359)
Author Response
- Although the main focus of the manuscript is on neuroimaging techniques, the authors include in their work a balanced load of pathophysiology and molecular neuropathology references related to WMHs, lacunar infarctions and dementia. In this regard, this reviewer considers that it would be interesting to add recent knowledge extracellular vesicles regarding WMHs and the vascular component of dementia and AD (see the following PUBMED identifiers PMID: 30629763; PMID: 32384937; PMID: 33098484;PMID: 29725359)
Response:We would like to thank the reviewer for evaluating our manuscript and for his/her comments. Following the reviewer’s suggestion, we have reviewed the aforementioned four studies, which are very interesting concerning the role of EVs. However, please note that this work reviewed the status of research on the features of WMHs related to VCI, as well as mediators associated with WMHs and VCI. Vascular disease increases the risk of AD and lower the threshold for AD diagnosis. VCI and AD have a mutually promoting relationship. Previous studies have shown that proteome specific changes in EVs and changes in the biogenic origin of EVs are closely related to cognitive impairment. However, their roles in the occurrence and development of WMHs and the related VCIs should be further examined. We believe that whether patients with WMHs have clinical symptoms of cognitive impairment may be observed in changes in the proteome of EVs, which may provide a new direction for our prognosis and treatment. This part has been added to the subsection named “Risk factors linking WMHs to VCI.” Especially, this finding is considered a possible factor of the occurrence and development of cognitive impairment and would be a direction of our future research. The added part is as follows:
“Recent findings have shown that cells in the CNS can release extracellular vesicles (EVs), which are a series of tiny double-membrane structures (20–2000 nm) with mem-brane-related interaction regions [61] and filled with different molecular materials (including proteins, lipids, DNA, and various RNA species). EVs targets nerve cells and blood vessel cells through a very dynamic and adaptable cell-cell communication mechanism, and changes the function of cells through the delivery of substances, thereby participating in the process of mediating neurovascular regeneration and remodeling, anti-apoptosis, and anti-inflammation and other processes [62]. Increased evidence has shown that proteome specific changes in EVs and these changes in the biogenic origin of EVs are closely related to the occurrence and development of cognitive impairment, blood perfusion, and cell apoptosis [63,64]. However, their roles in the occurrence and development of WMHs and their biomarker potential in predicting VCI risks or as a therapeutic approach for VCI in WMH patients are promising and should be further examined.” (Lines 283-295)
